# Venetoclax–Rituximab and Emerging Treatment Strategies After c-BTKi Exposure in Relapsed/Refractory CLL: A Real-World Cohort and Literature Overview

**DOI:** 10.3390/cancers17193159

**Published:** 2025-09-29

**Authors:** Maria Dimou, Rodanthi Fioretzaki, Calliope Zerzi, Eliana Konstantinou, John V. Asimakopoulos, Maria Arapaki, Alexia Piperidou, Alexandros Machairas, Anastasia Kopsaftopoulou, Athanasios Liaskas, Aikaterini Bitsani, Marina Belia, Fotios Panitsas, Aikaterini Benekou, Panagiota Petsa, Eleni Plata, Panagiotis Tsaftaridis, Marina Siakantaris, Theodoros P. Vassilakopoulos, Panayiotis Panayiotidis, Maria K. Angelopoulou

**Affiliations:** Department of Haematology and Bone Marrow Transplantation Unit, National and Kapodistrian University of Athens, Laikon General Hospital, 11527 Athens, Greece

**Keywords:** relapsed/refractory chronic lymphocytic leukemia, BTK inhibitors, real-world data, pirtobrutinib

## Abstract

**Simple Summary:**

Patients with chronic lymphocytic leukemia (CLL) who relapse after treatment with covalent Bruton tyrosine kinase inhibitors (c-BTKis) represent a growing and clinically challenging group, for whom evidence-based treatment options are still limited. Venetoclax combined with rituximab (VR) is an established fixed-duration therapy, but data on its use in c-BTKi-pretreated patients are scarce. In this retrospective study, we present real-world outcomes of VR in relapsed/refractory CLL, including patients previously exposed to c-BTKi. In parallel, we provide a systematic overview of available evidence from real-world studies and clinical trials of emerging therapies. Notably, most published real-world data on VR after c-BTKi exposure are still restricted to small cohorts or meeting abstracts. Our findings show that VR remains effective and safe even in this difficult-to-treat population and help place VR in context with novel agents such as non-covalent BTK inhibitors, thereby informing therapeutic decisions in daily practice.

**Abstract:**

Background: Fixed-duration venetoclax plus rituximab (VR) is a standard therapy for relapsed/refractory (R/R) chronic lymphocytic leukemia (CLL). However, evidence supporting its use after covalent BTK inhibitor (c-BTKi) therapy is scarce in clinical trials and limited in real-world settings. Objectives: To assess the efficacy and safety of VR in a real-world cohort of patients with R/R CLL, including cBTKi-pretreated individuals, and to contextualize outcomes alongside published real-world studies and registrational trials of alternative therapies. Methods: We retrospectively analyzed 37 patients with R/R CLL treated with VR at our center between April 2018 and November 2024. Baseline characteristics, treatment responses, minimal residual disease (MRD), and adverse events were recorded. Survival was estimated using the Kaplan–Meier method. A structured review of relevant real-world evidence and pirtobrutinib clinical trials was also conducted. Results: Median age was 67 years; 35.1% had prior cBTKi exposure. The overall response rate (ORR) was 91.7% (22/24 evaluable patients), with 66.7% achieving complete remission (CR). Among evaluable c-BTKi-pretreated patients, the ORR was 87.5% (7/8) and the CR rate was 62.5%. Undetectable MRD (uMRD) rates were 78.6% in peripheral blood and 71.4% in bone marrow. Thirty-month progression-free survival (PFS), time to next treatment (TTNT), and overall survival (OS) were >90% for the whole cohort and for c-BTKi-pretreated patients. The most frequent adverse event was neutropenia grade ≥ 3, especially during combination therapy, which is easily managed with GCSF support. Conclusions: Our real-world evidence shows that VR is an effective and well-tolerated option even after c-BTKi therapy in R/R CLL. These data complement evidence from emerging therapies and inform post-c-BTKi treatment selection in clinical practice.

## 1. Introduction

Chronic lymphocytic leukemia (CLL) is a heterogeneous lymphoproliferative disorder characterized by the progressive accumulation of mature B lymphocytes [1]. It remains the most prevalent form of adult leukemia in Western countries [2]. Over the past decade, the therapeutic landscape of CLL has been transformed with the advent of targeted agents, notably Bruton tyrosine kinase inhibitors (BTKis) and the BCL2 inhibitor venetoclax, which have significantly improved patient outcomes compared to traditional chemoimmunotherapy [3,4,5,6,7].

Venetoclax, a selective BCL2 inhibitor, in combination with rituximab (VR), has demonstrated robust efficacy and durable remissions in patients with relapsed/refractory (R/R) CLL, as demonstrated by the pivotal MURANO trial [8]. In this phase 3, randomized, controlled study, VR was compared to bendamustine plus rituximab (BR) in previously treated CLL patients. VR achieved a significantly superior progression-free survival (PFS) at 2 years (84.9% vs. 36.3%, HR 0.17; *p* < 0.001) and improved overall survival (OS) compared to BR. Additionally, VR therapy resulted in higher rates of undetectable minimal residual disease (uMRD) in peripheral blood (62.4% vs. 13.3%) after the completion of treatment. These findings established VR as a fixed-duration, chemotherapy-free regimen for R/R CLL patients. 

It should be noted that the vast majority of patients enrolled in the MURANO trial were BTKi-naïve, thus limiting the applicability of the above findings to the growing population of patients previously exposed to covalent BTKi (c-BTKi) therapies such as ibrutinib or the second-generation c-BTKis acalabrutinib and zanubrutinib [8,9,10]. Given the increasing use of c-BTKi as a frontline therapy, there is a critical need to understand the effectiveness and safety of VR in the post-BTKi setting. Real-world data (RWD), which captures outcomes in more diverse and less selected patient populations, is essential to inform clinical decision-making [11,12,13].

In this study we report real-world outcomes of VR in patients with R/R CLL, including a clinically relevant subset previously exposed to c-BTKi. In parallel, we provide a structured synthesis of published real-world studies and pivotal trial data of alternative therapies to place our findings on VR after c-BTKi exposure in the context of current treatment options.

## 2. Patients and Methods

This retrospective, single-center study included 37 consecutive patients with symptomatic R/R CLL requiring treatment [14] who initiated VR treatment between April 2018 and November 2024. Baseline patient and disease characteristics as well as efficacy and safety data for VR treatment were retrospectively collected from patients’ medical records. The treatment schedule followed the MURANO trial treatment design [8]: Venetoclax was administered orally with a standard 5-week ramp-up period to a target dose of 400 mg daily. Rituximab was administered at the dose of 375 mg/m^2^ intravenously on day 1 of the 1st cycle and 500 mg/m^2^ on day 1 of cycles 2–6. Venetoclax was continued for up to 24 cycles unless discontinued earlier for reasons of toxicity or disease progression. The scheduled duration of each cycle was 28 days. Responses at the end of treatment (EOT) or at any other time point as clinically indicated were assessed according to iwCLL 2018 criteria [14]. MRD was assessed by 8-color flow cytometry (FC) with a sensitivity of 10^™4^. [Kaluza™ Flow Analysis Software (Beckman Coulter, Brea, CA, USA)] European Research Initiative for CLL (ERIC) guidelines were used for FC MRD detection [15]. Adverse events were graded according to CTCAE v 5.0 [16]. Tumor lysis syndrome (TLS) prophylaxis was implemented according to standard operating procedures of our Department and in accordance with the summary of venetoclax product characteristics [17]. The primary endpoint was progression-free survival (PFS). Secondary endpoints were overall response rate (ORR), complete remission (CR) rate, MRD negativity, time to next treatment (TTNT), overall survival (OS), and safety. PFS was defined as the time from treatment initiation to disease progression or death; OS was defined as the time from treatment initiation to death from any cause; TTNT was defined as the time from treatment initiation to the start of next line of therapy.

Statistical analyses were performed using the SPSS software (version 29.0, IBM Corp., Armonk, NY, USA) [18]. The Kaplan–Meier methodology was used to estimate progression-free survival (PFS), overall survival (OS), and time to next treatment (TTNT). Median follow-up was estimated using the reverse Kaplan–Meier method. Median survival times with 95% confidence intervals (CIs) were reported when estimable, and survival probabilities at prespecified landmarks (12, 24, and 30 months) were calculated with standard errors. Survival distributions were compared between relevant subgroups (BTKi-naïve vs. c-BTKi-pretreated) using the log-rank test. Response rates and MRD negativity were summarized descriptively, and exact 95% CIs were provided for small subgroups (the Clopper–Pearson method).

## 3. Results

Thirty-seven (37) consecutive patients with R/R CLL initiated VR at our Center between April 2018 and November 2024. Baseline patient characteristics are shown in Table 1. The median follow-up time was 29.1 months (range: 2.3–80.9). The final data collection date for the present analysis was 20 February 2025.

At analysis, 24 patients were off therapy: 19 had completed the planned 24 cycles of VR and 5 had discontinued early (4 for adverse events, 1 for disease progression). Early discontinuations were due to newly diagnosed pancreatic cancer after 19 cycles, COVID-19 pneumonia after 17 cycles, severe pancytopenia after 8 cycles, persistent grade 4 neutropenia during the 1st venetoclax ramp-up dose, and Richter transformation in the 1st month. Among these 24 patients, responses were assessed per iwCLL 2018 using CT and bone marrow biopsy: in 22, the disease responded, including 16 complete remissions (CRs); the remaining 6 had partial remission (PR) driven solely by isolated, low-volume residual lymphadenopathy (>1.5 cm short axis, all <3 cm). Among 13 patients previously treated with a c-BTKi, 8 were evaluable at the final collection date; 7/8 achieved an objective response (ORR 87.5%, 95% CI 47.3–99.7), including 5/8 complete responses (CR 62.5%, 95% CI 24.5–91.5) and 2 partial responses. The median treatment-free interval among the 24 off-therapy patients was 17.5 months (range: 0–56.5). All other patients were on therapy at data cut-off.

In four patients, CLL progressed after a median time of 43.6 (1–62.6) months of VR initiation; one on treatment due to Richter transformation and three after the EOT. Among these patients with CLL progression, two had received one prior treatment line and two had received three or four lines, while two had prior cBTKi exposure. IGHV status was available in two patients and was mutated in both; *TP53* aberrations were present in two, and a complex karyotype (≥5 abnormalities) was present in one. One progression occurred on-treatment as Richter transformation in month 1 (death after one cycle of R-CHOP). For the remaining three cases, CLL progressed after VR completion; two patients started acalabrutinib and were in remission at data cut-off, while one remained off treatment.

PFS, TTNT, and OS for the whole cohort and separately for BTKi-naïve and BTKi-pretreated patients are shown in Figure 1. At a median follow-up of 29.1 months, the median PFS, OS, and TTNT were not reached (NR) in the overall cohort as well as in the BTKi-naïve and c-BTKi-pretreated subgroups. The Kaplan–Meier estimated 30-month outcomes for the entire cohort were 88.1% for PFS (95% CI: 75.0–98.0), 89.1% for OS (95% CI: 73.0–100), and 89.2% for TTNT (95% CI: 74.0–100). In BTKi-naïve patients, the corresponding 30-month rates were 86.1% (95% CI: 68.0–100), 87.5% (95% CI: 65.0–100), and 87.5% (95% CI: 65.0–100), respectively, while in c-BTKi–pretreated patients, they were 92.3% (95% CI: 78.0–100), 68.8% (95% CI: 28.0–100), and 46.2% (95% CI: 0–100). These estimates should be interpreted with caution given the small number of patients at risk beyond 24 months, particularly in the c-BTKi subgroup.

Among the 12 patients with unmutated IGHV status, the median PFS was not reached. In the *TP53*-aberrant subgroup (*n* = 4), median PFS was 30.2 months (95% CI, 9.3–51.1); two progression events were observed: one Richter transformation during the first cycle and one CLL progression after EOT. Among the 19 patients who completed VR treatment, MRD assessment at EOT was available for 14 of them. Undetectable MRD (uMRD) was achieved in 78.6% of peripheral blood samples, PB (in 11 patients with 9 of them in CR and 2 in PR), and in 71.4% of bone marrow samples, BM. 

The predominant adverse event associated with VR was neutropenia (all grades *n* = 16/37, 43.2%; grade ≥ 3, *n* = 12/37, 32.4%), whereas anemia grade 2 was observed in two patients (2.7%). All grade ≥ 3 neutropenia occurred early during the combination phase (cycles 2–5). In 9 of 12 cases, neutropenia persisted to EOT and required regular pre-emptive G-CSF, typically 1–2 times per week. No febrile neutropenia episode was noticed. Grade 4 thrombocytopenia developed in two patients (2.7%). Venetoclax dose modifications for hematologic toxicity were required in 16 patients (43.2%), including dose reductions in 4 (10.8%) and permanent discontinuation in 2 (5.4%). Respiratory infections occurred in five cases (13.5%), four of which were SARS-CoV-2 pneumonias. All SARS-CoV2 cases were hopsitalized without fatal outcome. One patient experienced grade 2 diarrhea (2.7%). There was no case of clinical or laboratory TLS in our cohort. 

## 4. Discussion

The optimal treatment strategy for R/R CLL patients previously exposed to c-BTKi remains a clinically relevant challenge [8,9,10]. While venetoclax-based combinations—particularly venetoclax plus rituximab (VR)—are recommended in this setting [9,19,20], data on their efficacy after c-BTKi therapy within randomized clinical trials are scarce [8]. Notably, the pivotal MURANO trial, which established the role of fixed-duration VR, included very few c-BTKi-pretreated patients—not by design, but rather because c-BTK inhibitors were not widely used at the time of patient enrollment. As a result, the efficacy of VR after prior c-BTKi exposure must be evaluated primarily through RWD. Although RWD have begun to address this evidence gap, only a handful of studies have specifically evaluated the effectiveness of fixed-duration VR in c-BTKi-pretreated patients, and most are limited by small sample sizes, retrospective design, or lack of stratified subgroup analysis. In particular, only three studies of the few available relevant studies have been fully published [21,22,23], with all others being only available in abstract form [24,25,26].

In 2024, Saburi et al. reported a small Japanese retrospective study of nine patients treated with VR, six of whom were c-BTKi-pretreated. The ORR was 88%, and 3-year PFS and OS rates were 83.3% [21]. The REVEAL study, a prospective, multicenter observational study conducted in Israel, included 144 R/R CLL patients, 72% (103) of whom received VR. BTKi exposure was reported in 51.4% of patients, and the ORR reached 90%, with a uMRD rate of 73% and estimated 2-year PFS and OS of 66% and 78%, respectively. However, outcomes for BTKi-exposed patients treated specifically with VR were not reported [24]. The AREVEDECY study from Belgium included 117 R/R CLL patients, 82 of whom received VR. The ORR in the VR group was 91%, but BTKi exposure within this group was not specified, precluding meaningful interpretation for this subgroup [25]. 

Only three recent real-world studies have clearly reported outcomes of VR in BTKi-pretreated patients [22,23,26]. The CORE study included 64 such patients, with an ORR of 71.4% and median PFS and TTNT of 39.5 and 37.4 months, respectively. MRD and CR data were not reported [22]. Additionally, an Australian real-world study published in 2024 included 32 BTKi-pretreated patients who had received fixed-duration VR [23]. The ORR was 81%, the CR rate was 19%, and MRD negativity (PB or BM) was observed in 70% of cases. Median PFS and OS were 25.9 and 46.1 months, respectively. Finally, in a recent prospective non-interventional real-world study conducted in Austria, Germany, and Switzerland, including 86 BTKi-exposed CLL patients, of whom 28 (33%) received venetoclax–rituximab, the VR cohort achieved a 100% overall response rate with a 2-year PFS of 72.9%, and no new safety signals were observed [26]. 

In this context, our real-world cohort contributes additional insights into the efficacy of the VR regimen. Among our 37 patients treated uniformly with VR, 13 (35.1%) had prior c-BTKi exposure. In this subgroup, the ORR was 91.7% and CR rate was 66.7%. The 30-month PFS, TTNT, and OS were all above 90%. These outcomes compare favorably with data from the CORE registry, as well as from the Australian and Austrian–German–Swiss cohorts, further supporting the role of fixed-duration VR after cBTKi therapy.

The data reported and analyzed here is particularly relevant in the context of the availability of non-covalent BTK inhibitors such as pirtobrutinib, which has been recently approved in Europe for R/R CLL patients who have received c-BTKi [27]. In the challenging post-c-BTKi setting, the clinically relevant treatment landscape essentially comprises fixed-duration VR and, more recently, pirtobrutinib. Other options such as venetoclax monotherapy, PI3K inhibitors, or re-treatment with a covalent BTKi have limited or no role in this context, and alternative combinations remain off-label [8,9,10]. Consequently, the management of a CLL patient after c-BTKi therapy poses a difficult question for every hematologist. The phase 1/2 BRUIN study evaluated pirtobrutinib in patients with R/R CLL, the majority of whom were heavily pretreated. Among 247 patients enrolled, 154 (62.3%) had received prior c-BTKi therapy and were naïve to BCL2 inhibitors. In this efficacy population, pirtobrutinib resulted in an ORR of 72%, increasing to 82% with the inclusion of partial responses with lymphocytosis, and a median PFS of 23.0 months [28]. In the phase 3 BRUIN CLL-321 trial (*n* = 247), all patients had prior cBTKi exposure, and ~50% were BCL2i-naïve. Pirtobrutinib significantly improved PFS (14.0 vs. 8.7 mo; HR 0.54; *p* = 0.0002) and TTNT (24.0 vs. 10.9 mo; HR 0.37) vs. the investigator’s choice of idelalisib–rituximab or bendamustine–rituximab, with fewer grade ≥ 3 AEs and discontinuations due to toxicity [29].

Table 2 provides a synthesis of outcomes from selected real-world studies considered most representative of fixed-duration VR use in c-BTKi-pretreated patients (including our current cohort), together with key results from the registrational trials of pirtobrutinib in the same clinical setting. While the BRUIN studies (both the phase 1/2 and the randomized BRUIN CLL-321) were well-designed and rigorously conducted prospective trials, the data on VR are derived from real-world observational studies with inherent limitations, including retrospective bias, lack of randomization, and heterogeneity in patient monitoring and assessment. Any comparison between these two treatment strategies must therefore be interpreted with caution, acknowledging that it is indirect and spans different methodologies, patient populations, and levels of data control. 

Taken together, these findings underscore the therapeutic relevance of fixed-duration venetoclax–rituximab in both c-BTKi-pretreated and BTKi-naïve R/R CLL. Among evaluable patients (24), high efficacy was observed, with an overall response rate of 91.7% in the entire cohort (22/24) and 87.5% (7/8) in the c-BTKi-pretreated subgroup. At 30 months, KM-estimated OS, TTNT, and PFS for the overall cohort were 97.3%, 97.3%, and 93.6%, respectively, while undetectable MRD rates at EOT were 78.6% in PB and 71.4% in BM. Early and often persistent grade ≥ 3 neutropenia was effectively mitigated with G-CSF and dose modifications, ensuring treatment continuity. 

A practical advantage of VR is its fixed-duration design, which provides a treatment-free interval and preserves future options at relapse, including re-treatment with venetoclax or venetoclax-based salvage combinations in selected patients [30,31]. Although detailed quantitative data on the efficacy of venetoclax after pirtobrutinib are limited, real-world clinical experience indicates that venetoclax-based regimens remain a viable option in this setting [32]. In practice, it seems that venetoclax can be sequenced either before or after pirtobrutinib, depending on prior exposures and patient-specific factors.

Our study has certain limitations. The cohort size was modest, and subgroup analyses, particularly among c-BTKi-pretreated patients, were based on small numbers at risk, resulting in wide confidence intervals and reduced precision of long-term estimates. Follow-up was heterogeneous, reflecting consecutive inclusion over several years, yet the median duration of nearly 30 months provides a mature dataset for landmark survival estimates. The retrospective, single-center design may introduce bias, but it also ensured uniform treatment and detailed longitudinal data capture. Finally, comparisons with pirtobrutinib data should be viewed as indirect, since those derive from prospective clinical trials, whereas our findings represent real-world experience; taken together, however, both sources offer complementary insights in the challenging post-cBTKi setting. Despite these limitations, this study provides one of the more mature single-center real-world datasets supporting the efficacy and safety of fixed-duration VR in R/R CLL, including cases after c-BTKi exposure.

## 5. Conclusions

Our real-world cohort, uniformly treated with venetoclax–rituximab, showed high efficacy and tolerability and adds to the available evidence supporting its role in everyday practice, even for patients previously exposed to covalent BTKi. The recent approval of pirtobrutinib has introduced another effective option for this group, based on data from well designed prospective clinical trials with heavily pretreated patients, and it broadens the therapeutic landscape. Although direct comparisons between studies should be made with caution, current data suggest that both approaches can provide meaningful benefit, with treatment choice depending on patient characteristics and preferences, previous therapy, and tolerability. Head-to-head studies and biomarker-driven approaches, if feasible, could clarify sequencing and guide more personalized care in R/R CLL after BTKi.

## Figures and Tables

**Figure 1 cancers-17-03159-f001:**
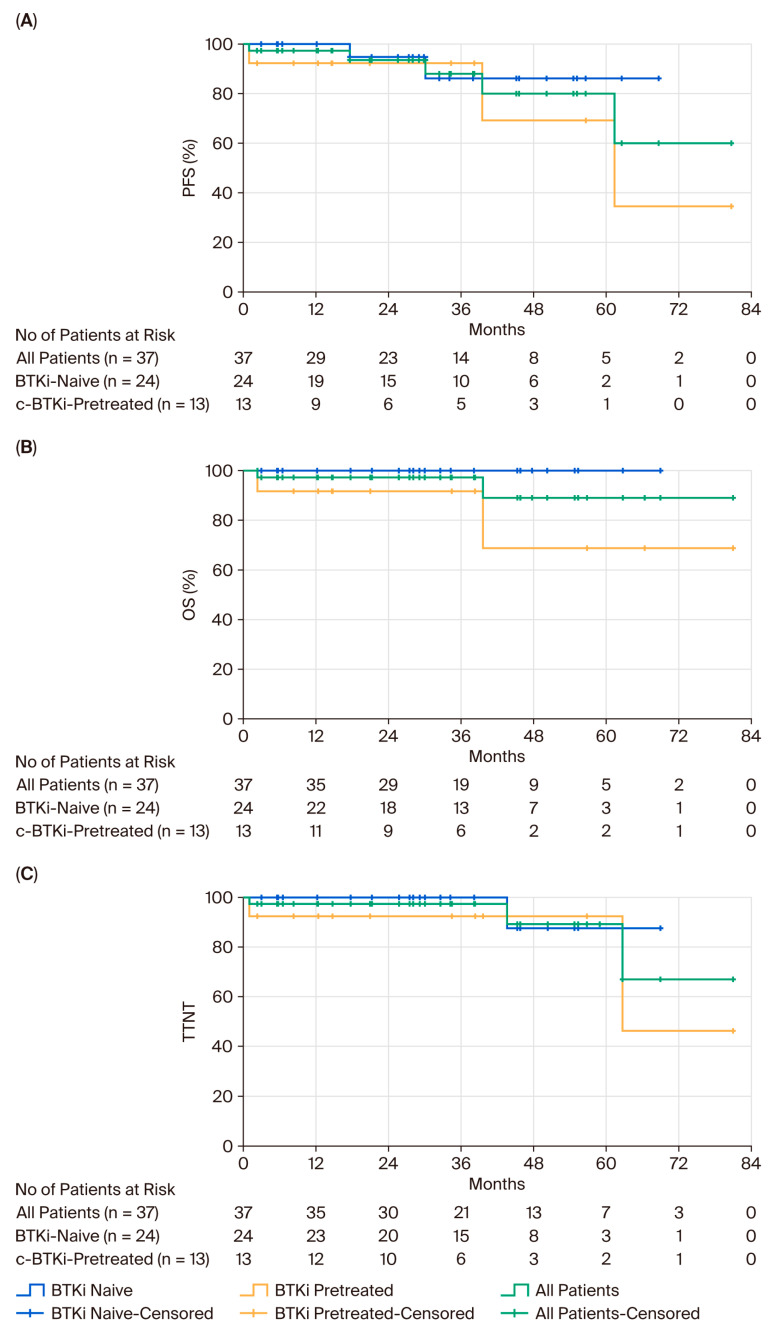
(**A**). progression-free survival (PFS), (**B**). overall survival (OS), and (**C**). time to next treatment (TTNT) for the whole cohort, the BTKi-naïve patients, and the BTKi-pretreated patients.

**Table 1 cancers-17-03159-t001:** Patient characteristics (*n* = 37).

Male	27 (73%)
Median age (range)	67 (44–89)
Median prior trm lines (range)	1 (1–4)
Pts with ≥2 prior trm lines	11 (29.7%)
Time since last trm, mo (range)	27 (0–128)
Prior CIT	30 (81.1%)
Prior BTKi	13 (35.1%) *
Reason for BTKi discontinuation	
disease progression	8 (61.5%)
adverse event	5 (38.5%)
IGHV status, unmutated (*n* = 23)	12 (52.1%)
*TP53* aberrations [del(17p) and/or *TP53* mutations] (*n* = 26)	4 (15.8%)
Complex karyotype (≥5 CA) (*n* = 29)	4 (13.8%)

Abbreviations: trm: treatment, pts: patients, mo: months CIT: chemoimmunotherapy, BTKi: Bruton tyrosine kinase inhibitor, IGHV: immunoglobulin heavy chain variable region, CA: chromosomal aberrations. *: all had received continuous ibrutinib; 1 had received both continuous ibrutinib and continuous acalabrutinib.

**Table 2 cancers-17-03159-t002:** Summary of VR (from RWD) and pirtobrutinib outcome (from clinical trials) in BCL2-naïve CLL patients after c-BTKi exposure [22,23,26,27,28].

Study	*n*	TRM	Age, Median (Range)	Median FU, mo (Range)	Prior Trm Lines, Median (Range)	Prior c-BTKi, *n* (%)	Reason for c-BTKi Discontinuation, (%)	ORR, %	CR Rate, %	PFS	TTNT	OS
Current cohort	37 ^#^	VR	67 (44–89) ^#^	29.1 (2.3–80.9) ^#^	1 (1–4) ^#^	13 (35.1)	PD: 8 (61.5) AE: 5 (38)	87.5	62.5	92.3% @30 mo	92.3% @30 mo	91.7% @30 mo
CORE study [22]	64	VR	NR	NR	NR	64 (100)	PD: 43 AE: 37	71.4	NR	39.5 mo (median)	37.4 mo (median)	NR
Australian RWD [23]	32	VR	70.5 (49–84)	20.6 (<1–58.6)	2 (1–5)	32 (100)	PD: 25 (78) AE: 7 (22)	81	19	25.9 mo (median)	NR	46.1 mo (median)
Austrian–German–Swiss RWD [26]	28	VR	59 (NR)	23 (NR)	3 (1–10)	28 (100)	PD: 10 (35.7) AE: 14 (50)	100	54	72.9% @ 24 mo	NR	76.6% @ 24 mo
Phase 1/2 BRUIN study [27]	154	Pirto	69 (36–87)	27.5 (22.2–31.6)	4 (1–11)	154 (100)	PD *: 76.9% AE *: 23.1%	83.1	5	23 mo (median)	NR	NE
BRUIN 321 Phase 3 study [28]	59	Pirto	67 (42–90)	NR	3 (1–13)	59 (100)	PD *: 71% AE *: 17%	NR	NR	NR	29.5 mo	NR

VR: venetoclax–rituximab, RWD: real-world data, CLL: chronic lymphocytic leukemia, c-BTKi: covalent Bruton tyrosine kinase inhibitor, *n*: number, trm: treatment, FU: follow-up, mo: months, ORR: overall response rate, CR: complete remission, Pirto: pirtobrutinib, PFS: progression-free survival, TTNT: time to next treatment, OS: overall survival, PD: progression of disease, AE: adverse event, pts: patients, @: at, NR: not reported, NE: not evaluable, all pts: the sum of patients enrolled in the study (all venetoclax-treated patients, or all c-BTKi. ^#^: all patients, *: all c-BTKi-treated patients irrespective of prior BCL2 exposure.

## Data Availability

The data presented in this study are available on reasonable request from the corresponding author. The data are not publicly available due to patient confidentiality.

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
