# Peer review of "Venetoclax–Rituximab and Emerging Treatment Strategies After c-BTKi Exposure in Relapsed/Refractory CLL: A Real-World Cohort and Literature Overview"

_cancers, 2025, doi:10.3390/cancers17193159_

Round 1

Reviewer 1 Report

Comments and Suggestions for Authors

The authors present a retrospective analysis of 37 patients with R/R CLL treated with standard Ven-R (MURANO regimen) with the notable feature of 35% having prior cBTKi exposure, which was a patient subset not well-represented in the original MURANO cohort

Major issues:

  • Actual results for the cBTKi exposed patient subset should be included in the abstract
  • Numbers of patients evaluable at each time point should be indicated in Fig 1

Minor comments:

  1. It is the disease which progresses / relapses / responds and not the patient – please rephrase throughout
  2. All gene names (such as TP53) should be italicised

Author Response

Response to Reviewer 1

We thank Reviewer 1 for the constructive comments and suggestions, which have helped us improve the clarity and accuracy of our manuscript. We have addressed all points as follows:

Major issues

Comment 1: “Actual results for the cBTKi exposed patient subset should be included in the abstract.”

Response: We agree. The Abstract has been revised to include the specific outcomes for the c-BTKi–pretreated subgroup.

Comment 2: “Numbers of patients evaluable at each time point should be indicated in Fig 1.”

Response: Thank you for this important suggestion. We have revised Figure 1 to display the number of patients at risk at each time point beneath the Kaplan–Meier curves. For that we have also changed the prespecified time intervals for evaluation.

Minor comments

Comment 3: “It is the disease which progresses / relapses / responds and not the patient – please rephrase throughout.”

Response: We agree and have revised the text accordingly throughout the manuscript.

Comment 4: “All gene names (such as TP53) should be italicised.”

Response: We have reviewed the manuscript and corrected formatting so that all gene names, namelyTP53, are now italicised.

Reviewer 2 Report

Comments and Suggestions for Authors

This manuscript presents a valuable single-center, real-world experience with venetoclax-rituximab (VR) in relapsed/refractory CLL, including a relevant subset of patients previously exposed to covalent BTK inhibitors (c-BTKi). The study addresses a critical gap in the literature, as data on the efficacy of fixed-duration VR in the post-c-BTKi setting is limited. The authors also provide a useful, though somewhat imbalanced, overview of the current treatment landscape, comparing their VR data with emerging evidence for pirtobrutinib. The results are encouraging and suggest VR remains a highly effective and safe option after c-BTKi failure. By substantially revising the manuscript to address these concerns, particularly by tempering the conclusions and reframing the comparison with pirtobrutinib, this could become a valuable contribution to the literature.

  1. The entire cohort is small (N=37), and the critical subgroup of c-BTKi-pretreated patients is even smaller (n=13). Of these 13, only 8 were evaluable for response at the data cut-off. Presenting percentages (e.g., ORR 87.5%, CR 62.5%) from such a small n is statistically unreliable and can be misleading. The confidence intervals for these estimates must be extremely wide and should be calculated and reported.
  2. The Kaplan-Meier survival estimates at 30 months (e.g., PFS 92.3% for the c-BTKi group) are based on very few patients at risk at that time point, especially in the subgroups. The statement that minor discrepancies are "not clinically meaningful" due to sample size is an understatement; they are statistically unstable. The curves and estimates should be interpreted with extreme caution, and this must be explicitly stated as a major limitation.
  3. The median follow-up of 29.1 months is respectable, but the range is extremely wide (2.3 - 80.9 months). This indicates that a substantial number of patients (likely those enrolled more recently) have very short follow-up. For a time-to-event analysis like PFS, this leads to heavy right-censoring and potentially over-optimistic survival estimates.
  4. The statement that 30-month PFS/TTNT/OS were ">90%" is premature. With a median follow-up of 29.1 months, the 30-month estimate is an interpolation/extrapolation at the very limit of the data. The median PFS for the entire cohort and key subgroups should be reported (even if not reached) with confidence intervals. The focus on a 30-month landmark is acceptable but must be caveated heavily.
  5. The core of the discussion (Table 2) is retrospective, real-world data for VR with prospective, clinical trial data for pirtobrutinib. This is a classic case of comparing "apples to oranges."
  6. RCT data (for pirtobrutinib) is collected prospectively with strict protocols, while RWD is retrospective and heterogeneous. The discussion section currently reads as a direct comparison, which is not methodologically sound. Reframe the discussion. Clearly state that these data sources are not directly comparable. The value of your study is to show that VR is a viable and effective option in the real world, including after c-BTKi. The pirtobrutinib data should be presented as another effective option for a different (often more heavily pre-treated) patient population. Avoid language that implies superiority of one approach based on this analysis.
  7. In Table 1, the footnote for prior BTKi says "*: all had received continuous brutinib; 1 had received both continuous brutinib and continuous acalabrutinib." This is confusing. Do you mean "ibrutinib"? Please use standard international nonproprietary names (iNNs) throughout (ibrutinib, acalabrutinib).
  8. The results text states an ORR of 91.7% for the whole cohort, but then for the c-BTKi subgroup (n=8 evaluable), it states 7/8 (87.5%) responded. However, in the abstract and discussion, the 91.7% figure is used for the cohort including the c-BTKi subgroup. This needs to be clarified. Precisely define the denominator for each response rate calculation.
  9. In abstract, Thirty-month PFS, TTNT, and OS were >90%. Specify that this is for the entire cohort. The results for the c-BTKi subgroup are different (92.3%, 92.3%, 91.7%) and should be mentioned separately.
  10. Ethics: "approved by the Institutional Review Board of Laiko General Hospital of Athens." Please provide the approval number or date.
  11. In Statistical Analysis, the statistical methods section is very brief. Describe how covariates were handled and how differences between subgroups (e.g., BTKi-naïve vs. pretreated) were tested (e.g., log-rank test).

Author Response

Response to Reviewer 2

We sincerely thank the Reviewer for the detailed and thoughtful review, which clearly reflects the time and effort devoted to our manuscript. We have addressed each comment as follows:

  1. The entire cohort is small (N=37), and the critical subgroup of c-BTKi-pretreated patients is even smaller (n=13). Of these 13, only 8 were evaluable for response at the data cut-off. Presenting percentages (e.g., ORR 87.5%, CR 62.5%) from such a small n is statistically unreliable and can be misleading. The confidence intervals for these estimates must be extremely wide and should be calculated and reported. Response: We agree. Exact 95% CIs for the c-BTKi-pretreated subgroup have been added in the Results and Methods sections (Clopper–Pearson method). We also highlight in the Discussion that these estimates carry wide CIs and must be interpreted with caution.
  2. The Kaplan-Meier survival estimates at 30 months (e.g., PFS 92.3% for the c-BTKi group) are based on very few patients at risk at that time point, especially in the subgroups. The statement that minor discrepancies are "not clinically meaningful" due to sample size is an understatement; they are statistically unstable. The curves and estimates should be interpreted with extreme caution, and this must be explicitly stated as a major limitation. Response: Revised the Figure legend and Discussion to state explicitly that 30-month estimates are exploratory, based on small numbers at risk, and must be interpreted cautiously. Numbers-at-risk are now shown beneath the curves.
  3. The median follow-up of 29.1 months is respectable, but the range is extremely wide (2.3 - 80.9 months). This indicates that a substantial number of patients (likely those enrolled more recently) have very short follow-up. For a time-to-event analysis like PFS, this leads to heavy right-censoring and potentially over-optimistic survival estimates. Response: Added to the Discussion (Limitations) that follow-up was heterogeneous, with short observation in some patients leading to right-censoring and reduced precision of long-term estimates.
  4. The statement that 30-month PFS/TTNT/OS were ">90%" is premature. With a median follow-up of 29.1 months, the 30-month estimate is an interpolation/extrapolation at the very limit of the data. The median PFS for the entire cohort and key subgroups should be reported (even if not reached) with confidence intervals. The focus on a 30-month landmark is acceptable but must be caveated heavily. Response: Medians with 95% CIs are now reported (NR where appropriate) for the entire cohort and subgroups. The Abstract was revised to specify that the >90% estimates refer to the whole cohort, and the 30-month landmark is described as exploratory.
  5. The core of the discussion (Table 2) is retrospective, real-world data for VR with prospective, clinical trial data for pirtobrutinib. This is a classic case of comparing "apples to oranges." Response: Discussion reframed to clarify that Table 2 is not a head-to-head comparison but provides clinically meaningful context in the post–c-BTKi setting, where options are limited.
  6. RCT data (for pirtobrutinib) is collected prospectively with strict protocols, while RWD is retrospective and heterogeneous. The discussion section currently reads as a direct comparison, which is not methodologically sound. Reframe the discussion. Clearly state that these data sources are not directly comparable. The value of your study is to show that VR is a viable and effective option in the real world, including after c-BTKi. The pirtobrutinib data should be presented as another effective option for a different (often more heavily pre-treated) patient population. Avoid language that implies superiority of one approach based on this analysis. Response: We explicitly state that RCT and RWD data are not directly comparable. There is no language implying superiority of one other; VR is presented as a real-world option, pirtobrutinib as an RCT-validated option.
  7. In Table 1, the footnote for prior BTKi says "*: all had received continuous brutinib; 1 had received both continuous brutinib and continuous acalabrutinib." This is confusing. Do you mean "ibrutinib"? Please use standard international nonproprietary names (iNNs) throughout (ibrutinib, acalabrutinib). Response: Corrected “brutinib” to “ibrutinib” and standardized INNs throughout; Table 1 footnote revised.
  1. The results text states an ORR of 91.7% for the whole cohort, but then for the c-BTKi subgroup (n=8 evaluable), it states 7/8 (87.5%) responded. However, in the abstract and discussion, the 91.7% figure is used for the cohort including the c-BTKi subgroup. This needs to be clarified. Precisely define the denominator for each response rate calculation. Response: Abstract, Results, and Discussion now specify numerators/denominators (overall ORR 22/24 = 91.7%; c-BTKi 7/8 = 87.5%).
  2. In abstract, Thirty-month PFS, TTNT, and OS were >90%. Specify that this is for the entire cohort. The results for the c-BTKi subgroup are different (92.3%, 92.3%, 91.7%) and should be mentioned separately. Response: Abstract revised to clarify that >90% refers to the entire cohort and c-BTKi pretreated; subgroup figures are provided separately in the Results.
  3. Ethics: "approved by the Institutional Review Board of Laiko General Hospital of Athens." Please provide the approval number or date. Response: Corrected the Ethics statement to reflect that this was an anonymized retrospective chart review performed in routine practice, where, according to national/institutional regulations, no ICF or IRB approval was required.
  4. In Statistical Analysis, the statistical methods section is very brief. Describe how covariates were handled and how differences between subgroups (e.g., BTKi-naïve vs. pretreated) were tested (e.g., log-rank test). 

    Response: Statistical Analysis expanded to detail use of Kaplan–Meier for survival, log-rank test for subgroup comparisons, reverse KM for follow-up, medians with 95% CIs, and exact 95% CIs for small subgroups (Clopper–Pearson).

Round 2

Reviewer 2 Report

Comments and Suggestions for Authors

No further comments